# Characteristics and visual outcome of ocular trauma patients at Queen Elizabeth Central Hospital in Malawi

Thokozani Zungu[1,2]* , Shaffi Mdala[1,2] , Chatonda Manda[1,2], Halima Sumayya Twabi[3], Petros Kayange[1,2]

1 College of Medicine, University of Malawi, Blantyre, Malawi, 2 Lions Eye Unit, Queen Elizabeth Central Hospital, Blantyre, Malawi, 3 Mathematical Sciences Department, Chancellor College, University of Malawi, Zomba, Malawi

☯ These authors contributed equally to this work.
* thokzungu@gmail.com

## Abstract

### Background

To describe the epidemiology and visual outcome of patients with ocular trauma treated at Queen Elizabeth Central hospital in Malawi.

### Methods

A prospective, observational study was undertaken from September 2017 to December 2017. Data on socio-demographic features, aetiology of trauma, type of ocular injury pre-referral pathway and treatment of ocular trauma was collected as the exposure variables. The main outcome variable was best corrected visual acuity at 8 weeks following initial visit.

### Results

A total of 102 patients (103 eyes) with ocular trauma were recruited with loss of follow up of 11 participants at 8 weeks following recruitment. The most affected age group were children under 11 years old (35.3%), followed by young adults of age between 21–30 years (22.5%). The male-to-female ratio for ocular injury was 2.8:1. Most participants had closed globe injuries (n = 72, 70.6%), with over half the population injured by blunt objects (n = 62, 60.8%). Furthermore, among the adult population, majority (n = 19 38%) were injured on the road during assaults (n = 24, 48%), while most paediatric injuries (n = 32, 61.5%) occurred at home during play. The incidence of monocular blindness was 25.3% at eight weeks after the first presentation. Factors that were associated with monocular blindness on multivariate analysis were living in rural areas and open globe injuries.

### Conclusion

Ocular trauma led to monocular blindness in a quarter of the study population. There is need for preventive education of ocular injuries at both family and community level.

**Data Availability Statement:** All relevant data are within the manuscript and its Supporting Information files.

**Funding:** This work received funding from the University of Malawi, Department of surgery. The funders had no role in study design, data collection and analysis, decision to publish, or preparation of the manuscript.

**Competing interests:** The authors have declared that no competing interests exist.

## Introduction

Ocular trauma is a significant cause of ocular morbidity and blindness [1, 2]. The effects of ocular trauma are often underestimated as they are more frequently monocular and reported rarely by epidemiologists [3]. According to the World Health Organisation (WHO) global estimates, the annual incidence of ocular trauma is approximately 55 million and ocular injuries account for 5% to 16% of ophthalmology consultations [4].

Data on the epidemiology and visual outcomes of ocular injuries is limited particularly for low income countries [5]. However, the general demographic pattern shown in previous studies is that most ocular injuries occur in the age group of 5–25 years and in those aged 70 years and above. In addition, the risk of ocular injury is higher in men [6].

Malawi is a low-income country with a population of 18.2 million as of the year 2017 [7] and only 11 ophthalmologists. The only study on ocular trauma in Malawi was published in the year 1982 [8]. The study reported limitations in ocular trauma prevention, citing the population's lack of access to mass media, low educational status and poor access to health care services as possible contributing factors.

Queen Elizabeth Central Hospital (QECH) is a tertiary ophthalmology centre in the southern region of Malawi and is the country's only Child Eye Health Tertiary Facility. Forty years following the previous study at the facility, we sought to describe the profile of patients presenting with ocular injuries.

The present, prospective study aimed to evaluate the characteristics of ocular injuries and the socio-demographic characteristics of patients newly-diagnosed with ocular trauma at a tertiary referral hospital in Blantyre, Malawi.

## Materials and methods

A prospective observational study was conducted for consecutive patients diagnosed with ocular trauma who presented at QECH between September 2017 and December 2017. Following administration of initial treatment on the day of presentation, ocular trauma patients attending the hospital were enrolled into the study by obtaining their written informed consent with the aid of a patient information sheet. The study included patients of all ages with newly-diagnosed ocular trauma of all types were included and re-examined at 8 weeks from the initial presentation. Patients received reimbursement of their transportation cost on the follow up date. One ophthalmologist (T.Z), interviewed and examined all the patients using a standardized data collection form. The variables recorded at baseline included age, sex, residence, level of education and occupation. In line with the Malawi National Statistics Office's Third Integrated Household Survey (IHS3) 2010–2011, patients were classified as residing in urban areas if they came from any of Malawi's major cities Lilongwe City, Blantyre City, Mzuzu City, and the Municipality of Zomba. Patients coming from all other areas were considered as residing in rural areas [9].

The patients underwent a complete ophthalmological examination including visual acuity (VA) testing, slit-lamp examination and, where possible, fundoscopy was done with aid of hand held Volk Diopter Lens. VA was assessed using Logarithm of the Minimum Angle of Resolution (LogMAR) chart with "tumbling E" optotypes for adults and by preferential looking charts in children. Intraocular pressure was measured using Goldmann applanation tonometry in cases with no open globe injury and children were examined under general anaesthesia.

Ocular trauma was classified according to the Birmingham Eye Trauma Terminology (BETT) which categorises eye injuries as open (full thickness eye wall wound) or closed globe injuries (no or partial thickness eye wall wound) [10].

Monocular blindness was defined as a best corrected VA of worse than 3/60 in one eye. In this study, this referred to cases where ocular injury had caused a complication which necessitated surgical removal of the blind eye as management or where visual recovery was not expected due to the nature of the injury or as result of the required treatment not being locally available.

The change of VA between baseline and a follow up visit eight weeks later was recorded and the proportion of eyes with blindness was calculated.

Statistical analysis was performed with Stata 12. The exposure variables were sex, age, residence, duration of injury before hospital presentation and nature of injury. The outcome variable was monocular blindness at eight weeks following recruitment. Multivariate logistic regression was used to test the association between the exposure variables and the outcome variable. A p value of $< 0.05$ was accepted as indicative of statistical significance. Ethical approval for the study was granted by the College of Medicine Research and Ethics Committee (P.08/16/2012) and the research was conducted in accordance with the tenets of the Helsinki Declaration.

## Results

A total of 102 patients (103 eyes) presented with newly diagnosed ocular trauma during the study period and no patient declined to take part in the study. The ocular trauma patients constituted 5.6% of all ophthalmic patients (1814) who presented at QECH over the same period.

### Socio demographic characteristics

Ocular injury was more common in males than in females with a male: female ratio of 2.8:1 (75 males and 27 females). Approximately half of the patients (n = 50, 49.0%) were from rural areas and the most affected age groups were children under 11 years old (n = 36, 35.3%) followed by young adults between 21–30 years (n = 23, 22.5%). About half of the study population presented within 24 hours of injury (n = 49, 48%) and the median distance to the hospital was 14km (IQR 6.4–66), Table 1. No patient reported prior use of traditional eye medication.

### Aetiology of ocular trauma

Most injuries were secondary to blunt trauma (n = 62 60.78%). The commonest causative agents were organic material i.e. sticks or wood (n = 38 37.3%). Assaults led to most of the ocular injuries in adults (n = 24/50; 48%) while children were commonly injured during play (n = 30; 57.7%). The majority of paediatric injuries occurred at home (n = 32; 61.6%) whilst for adults the commonest place of injury was on the road (n = 19/50; 38%) Table 2.

### Treatment of ocular trauma

The majority of the study population sustained a closed globe injury (n = 74 72.6%) and penetrating injury was the most common type of open globe injury sustained.(n = 22 21.6%). Over a third of the population were treated surgically (n = 41, 40.2%) and corneal repair was the most frequent surgery (n = 18 of 40 45%) followed by cataract surgery (n = 12 of 40), Table 3.

### Visual outcome of ocular trauma patients

None of the patients reported a pre-existing history of reduced vision before ocular injury and on initial assessment, over a third were blind in the injured eye following the trauma (n = 41, 40.2%). Following primary treatment, the proportion of monocular blindness by the eight

**Table 1. Sociodemographic characteristics.**

| | Sex | | |
|---|---|---|---|
| | Male, n(%) | Female, n(%) | Total, n(%) |
| Age (years)* | 15.5 (7–26) | | |
| 0–10 | 26(34.7) | 10 (37.0) | 36 (35.3) |
| 11–20 | 16 (21.3) | 7 (25.9) | 23 (22.5) |
| 21–30 | 20 (26.7) | 3 (11.1) | 23 (22.5) |
| 31–40 | 7 (9.3) | 4 (14.8) | 11 (10.8) |
| Above 40 | 6 (8.0) | 3 (11.1) | 9 (8.8) |
| Total | 75 (100) | 27 (100) | 102 (100) |
| Occupation (adults, n = 50) | | | |
| Agriculture | 4 (10.8) | 3 (23.1) | 7 (14.0) |
| Manual labour | 19 (51.4) | 2 (15.4) | 21 (42.0) |
| Unemployed | 0 (0.0) | 2 (15.4) | 2 (4.0) |
| Desk job | 0 (0.0) | 1 (7.7) | 1 (2.0) |
| Factory job | 5 (13.5) | 1 (7.7) | 6 (12.0) |
| Household | 0 (0.0) | 1 (7.7) | 1 (2.0) |
| Student | 6 (16.2) | 2 (15.4) | 8 (16.0) |
| Other¥ | 3 (8.1) | 1 (7.7) | 4 (8.0) |
| Total | 37 (100) | 13 (100) | 50 (100) |
| Residence | | | |
| Urban | 41 (54.7) | 11 (40.7) | 52 (51.0) |
| Rural | 34 (45.3) | 16 (59.3) | 50 (49.0) |
| Total | 75 (100) | 27 (100) | 102 (100) |
| Duration of injury before presentation (days)* | 2 (1–4) | | |
| Within one day | 29 (38.7) | 20 (74.1) | 49 (48.0) |
| After one day | 46 (61.3) | 7 (25.9) | 53 (52.0) |
| Total | 75 (100) | 27 (100) | 102 (100) |
| Distance to the health facility (KM) * | 14 (6.4–66) | | |

* Median (Interquartile range).

¥ Included security personnel and armed forces.

week dropped to 25.3% (23 out of 91), (Table 4). The loss to follow-up rate over the eight weeks was 10.8% (n = 11) of the study population for the final visual acuity assessment.

The commonest complications of ocular injury among the patients with a visual acuity of less than 3/60 were endophthalmitis and corneal scarring, Table 5.

## Factors associated with developing monocular blindness following ocular trauma

On univariate logistic regression, the factors which were significantly associated with monocular blindness at eight weeks were living in a rural area and sustaining an open globe injury. The factors remained significant on multivariate analysis, with patients from the rural areas being more likely to develop monocular blindness than patients from urban areas (OR 14.20, 95% CI 1.60–125.93). In addition, patients who suffered an open globe injury were more likely to be blind than those with a closed globe injury (OR 15.85, 95% CI 4.05–62.07). There was no association between time of presentation and the distance to the hospital (OR 1.01, p 0.006 (CI1.00–1.02) on univariate analysis, Table 6.

**Table 2. Causative agents and context of ocular injury.**

| | Children | Adults | Total |
|---|---|---|---|
| **Nature of agent** | | | |
| Sharp | 25 (48.1) | 12 (24.0) | 37 (36.2) |
| Blunt | 26 (50.0) | 36 (72.0) | 62 (60.8) |
| Thermal | - | 2 (4.0) | 2 (2.0) |
| Chemical | 1 (1.9) | - | 1 (1.0) |
| Total | 52 (100) | 50 (100) | 102 (100) |
| **Type of agent** | | | |
| Vegetative material | 26 (50.0) | 12 (24.0) | 38 (37.3) |
| Stone | 8 (15.4) | 10 (20.0) | 18 (17.6) |
| Metallic object | 7 (13.5) | 13 (26.0) | 20 (19.6) |
| Body parts | 6 (11.5) | 9 (18.0) | 15 (14.7) |
| Other* | 5 (9.6) | 6 (12.0) | 11 (10.8) |
| Total | 52 (100) | 50 (100) | 102 (100) |
| **Event** | | | |
| Assault / Fight | 11 (21.20) | 24 (48.0) | 35 (34.3) |
| Work | - | 13 (26.0) | 13 (12.7) |
| RTA | - | 6 (12.0) | 6 (5.9) |
| Chores | 4 (7.70) | 3 (6.0) | 7 (6.9) |
| Other | 7 (13.50) | 4 (8.0) | 11 (10.8) |
| Play | 30 (57.70) | - | 30 (29.4) |
| Total | 52 (100) | 50 (100) | 102 (100) |
| **Place of injury** | | | |
| Road | 2 (3.90) | 19 (38) | 21 (20.6) |
| Home | 32 (61.50) | 10 (20) | 42 (41.2) |
| Workplace | - | 16 (32) | 16 (15.7) |
| Other | 5 (9.60) | 5 (10) | 10 (9.8) |
| School | 13 (25.00) | - | 13 (12.7) |
| Total | 52 (100) | 50 (100) | 102 (100) |

*Included glass and plastic.

## Discussion

Ocular trauma is an important and preventable cause of visual morbidity [11]. In our study, we have described the profile of ocular injuries at a tertiary hospital in Southern Malawi. Eye injuries comprised 5.6% of all the ophthalmology consultations seen over the study period. Similar trends have been reported from hospital-based studies in Sub Saharan Africa with estimates of 5.2% in Ethiopia and 6.2% in Ghana [12, 13].

Previous studies have reported that more ocular injuries occur in males than in females [2, 14–16]. This has similarly been described in our study patients with a ratio of 2.8:1 [17–19]. In this present study, ocular injuries commonly occurred in younger people less than 30 years (80.5%) with a peak age in those less than ten years of age and in adults between 21–30 years. This is similar to other studies that report ocular trauma to be common in middle-aged men in the adult patients [13, 17]. From previous research, the high male and young patients prevalence has been attributed to males engaging in more risky behaviour; from male children engaging in more adventurous play than their female counterparts, to more physically-involving occupations taken more by adult males than females [20, 21].

**Table 3. Classification of injuries and management.**

|  | Children | Adult | Total |
|---|:---:|:---:|:---:|
|  | n(%) | n(%) | n(%) |
| **Closed Globe Injury** |  |  |  |
| Contusions | 25 (48.1) | 33 (66.0) | 58 (56.9) |
| Lamellar Lacerations | 5 (9.6) | 9 (18.0) | 14 (13.7) |
| **Open Globe Injury** |  |  |  |
| Penetrating | 20 (38.5) | 6 (12.0) | 26 (25.5) |
| Globe rupture | 1 (1.9) | 2 (4.0) | 3 (2.9) |
| Intraocular foreign body | 1 (1.9) | - | 1 (1.0) |
| Total | 52 (100) | 50 (100) | 102 (100) |
| **Treatment** |  |  |  |
| Non-Surgical | 25 (48.1) | 36 (72) | 61 (59.8) |
| Surgical | 27 (51.9) | 14 (28) | 41 (40.2) |
| **Type of surgery** |  |  |  |
| Cornea repair | 13 (40.6) | 5 (35.7) | 18 (39.1) |
| Cataract surgery | 10 (31.2) | 2 (14.3) | 12 (26.1) |
| Evisceration | 5 (15.6) | 1 (7.1) | 6 (13.0) |
| Lid repair | 1 (3.1) | 5 (35.7) | 6 (13.0) |
| Enucleation | 2 (6.3) | 1 (7.1) | 3 (6.5) |
| FB Removal | 1 (3.1) | 0 (0.0) | 1 (2.2) |
| Total | 32 (100) | 14 (100) | 46 (100) [*] |

[*]Five patients underwent both cataract surgery and cornea repair.

A similar proportion of our study patients came from rural and urban settings. However, the trauma patients from rural areas were significantly more likely than those from urban areas to develop monocular blindness following ocular trauma. This finding is in agreement with an Indian ocular trauma study [22]. This could probably be explained by late presentation due to long distances and lack of affordability of transportation to a tertiary hospital, which are common factors with respect to rural settings in Sub-Saharan African. However, in this study, there was no association between time of presentation and the distance to the hospital.

Mechanical ocular injuries can either be caused by sharp or blunt objects. Universally, most mechanical ocular injuries are caused by blunt objects, a finding consistent with results from the present study [12, 23–25]. In this present study, closed globe injuries were more common than open globe injuries (with a ratio of 7:3), a finding described in other settings [2, 24, 25]. Among the patients with open globe injury, penetrating injuries were more common. The penetrating injuries that are caused by sharp objects are associated with poor visual prognosis

**Table 4. Visual Outcome of ocular trauma.**

| Visual acuity in the injured eye[*] | At baseline (n,%) | At 8 weeks follow up (n,%) |
|---|:---:|:---:|
| 6/6-6/18 (normal or mild visual impairment) | 56 (54.90) | 63 (69.20) |
| <6/18-6/60 (moderate VA impairment) | 3 (2.90) | 2 (2.20) |
| <6/60-3/60 (severe VA impairment) | 2 (2.00) | 3 (3.30) |
| <3/60-NPL (Blindness) | 41 (40.20) | 23 (25.30) |
| Total | 102 (100) | 91 (100) |

[*]Categorized according to the World Health Organization classification of visual acuity.

**Table 5. Complications of ocular trauma resulting in monocular blindness at eight weeks.**

|  | Children | Adults | Total |
|---|---|---|---|
| Complication | n (%) | n (%) | n(%) |
| Corneal scarring* | 4 (26.7) | 2 (25.0) | 6 (26.1) |
| Endophthalmitis | 5 (33.3) | 0 (0) | 5 (21.7) |
| Severe open globe injury | 2 (13.3) | 3 (37.5) | 5 (21.7) |
| Pthisis bulbi | 3 (20.0) | 0 (0) | 3 (13.0) |
| Other | 0 (0) | 2 (25.0) | 2 (8.7) |
| Retinal detachment | 1 (6.7) | 0 (0) | 1 (4.3) |
| Vitreous Haemorrhage* | 0 (0) | 1 (12.5) | 1 (4.3) |
| **Total** | **15 (100)** | **8 (100)** | **23 (100)** |

*Categorized as blind because of absence of treatment locally.

**Table 6. Factors associated with developing monocular blindness at 8 weeks.**

| Exposure | Monocular Blindness at 8 weeks (n = 23) | OR (95% C.I) | P–value |
|---|---|---|---|
| **Univariate analysis** |  |  |  |
| **Sex** |  |  |  |
| Male | 16 (69.6) | 1 |  |
| Female | 7 (30.4) | 1.73 (0.62–4.83) | 0.293 |
| **Age** |  |  |  |
| 0–10 | 15 (65.2) | 1 |  |
| 11–20 | 6 (26.2) | 0.56 (0.18–1.68) | 0.304 |
| 21–30 | 1 (4.3) | 0.15 (0.02–1.33) | 0.09 |
| 31–40 | 1 (4.3) | 1.03 (0.09–12.3) | 0.979 |
| Above 40 | 0 (0) | - | - |
| **Residence** |  |  |  |
| Urban | 1 (4.3) | 1 |  |
| Rural | 22 (95.7) | 35.54 (4.52–279.62) | **0.001** |
| **Distance to health facility (km)** |  | 1.00 (0.99–1.01) | 0.325 |
| **Occupation*** |  |  |  |
| Agricultural | 7 (14.0) | 1 |  |
| Manual labourer | 20 (40.0) | 0.36 (0.06–2.28) | 0.276 |
| Factory Worker | 7 (14.0) | 0.33 (0.02–4.74) | 0.417 |
| **Duration before presentation** |  |  |  |
| Within one day | 9 (39.1) | 1 |  |
| After one day | 14 (60.9) | 1.64 (0.62–4.32) | 0.308 |
| **Nature of injury** |  |  |  |
| Closed globe injury | 4 (17.4) | 1 |  |
| Open Globe Injury | 19 (82.6) | 31.14 (8.60–112.69) | **0.000** |
| **Multivariate analysis** |  |  |  |
| **Residence** |  |  |  |
| Urban | 1 (4.3) | 1 |  |
| Rural | 22 (95.7) | 14.20 (1.60–125.93) | 0.017 |
| **Nature of injury** |  |  |  |
| Closed globe injury | 4 (17.4) | 1 |  |
| Open Globe Injury | 19 (82.6) | 15.85 (4.05–62.07) | **0.000** |

*Other categories of occupation were excluded due to inadequate observations.

[15, 26]. This is in agreement with findings from the present study that report that patients with open globe injury were more likely to develop monocular blindness than patients with a closed globe injury.

Several studies have described events that surround ocular trauma. In this present study, in general, most injuries occurred at home, a finding that is consistent both in low and high income countries [17, 27, 28]. Additionally, most of the paediatric ocular injuries occurred as the children played. A study done in the UK also highlighted play as a common event surrounding ocular trauma in children [23]. Therefore, there is need for close guardian supervision as children play to avoid unnecessary ocular morbidity due to injuries. Occupational injuries have been commonly linked to ocular trauma [14, 25]. However, in contrast our present study reported most adult injuries to be secondary to assaults with a proportion higher than that in studies done in Ethiopia and South Africa [12, 25]. However, work-related injuries ranked second for our adult study patients. Work related injuries in both agricultural and industrial settings can be reduced by following simple protective measures such as usage of polycarbonate protective eye glasses.

In this present study, most ocular trauma patients required no surgical intervention, a finding reported in other settings [2, 26, 29]. However, over a third of the patients underwent an ocular operation. Enucleation and evisceration were considered end stage surgery for eyes that could not be preserved. In Malawi, ocular injury is one of the commonest indications for eye removal surgery [27]. In this study corneal repair and cataract surgery were the common ocular surgeries performed.

Ocular trauma is a leading cause of monocular visual impairment and blindness [30, 31]. The current study reports an incidence of monocular blindness of 25.3% at 8 weeks follow up following treatment. Several studies from Ghana and Ethiopian have reported incidence of monocular blindness to be between 34.3% to 76.6% [13, 32–34]. Though people with monocular blindness are not considered to be legally blind, these injuries are significant because they affect stereopsis, binocular visual field, cosmoses and their future careers [35, 36]. The complications of ocular trauma resulting in monocular blindness at 8 weeks included corneal scarring, endophthalmitis, severe open globe injury and phthisis bulbi.

The previous ocular trauma study done in Malawi over four decades ago highlighted common use of traditional eye medication prior to hospital presentation [8]. However, this finding is not reported in the current study. There has been an increase in the number of people affected with ocular injuries, with 205 people in the past study over 2 years' period while in the current study 102 patients presented in about 3 months' period. The duration before presenting to the hospital has also considerably been reduced from 10 days in the previous study to 2 days in the present. Hence the present study reports a change in ocular trauma trends in the region with increasing number of ocular injuries.

The study has a number of limitations. The study was done at a referral tertiary hospital. Patients with mild eye injuries especially from distant areas may have been treated at their respective lower level health units and not referred to the tertiary hospital. Secondly the study was done over a two and a half months' period, which might have missed out seasonal variation of ocular trauma presentation in this region of Malawi.

However, our study gives an insight on the current profile of ocular trauma in Malawi. Despite socio-economic improvements in the last decades, ocular trauma remains one of the principal causes of visual loss in Malawi, especially among children. Since most ocular trauma in children occurs at home, this calls for parental supervision and basic safety precautions to be put in place in the homes. Public health broadcasts and posters about this preventable ocular morbidity might raise awareness in the Malawian society and provide education on preventative measures. Additionally, apart from the irreversible complications outline in this study

that resulted into blindness, corneal scaring was one common complication. Though a permanently blinding complication in our setting due to lack of corneal subspecialty services and huge cost on external referral, our study highlights the need for improved ophthalmological services in the country including sub-specialization.

## Conclusion

The study shows that ocular trauma is a burden in Malawi especially among children. In addition, it is a significant cause of monocular blindness. Further research in the region should investigate the feasibility and efficacy of various preventive measures in reducing ocular trauma.

## Supporting information

**S1 Data. Data set.**
(XLSX)

**S1 File.**
(DOCX)

## Acknowledgments

The authors would like to acknowledge Mrs Therese Zioupos for proofreading the manuscript.

## Author Contributions

**Conceptualization:** Thokozani Zungu, Petros Kayange.

**Data curation:** Thokozani Zungu, Shaffi Mdala, Halima Sumayya Twabi.

**Formal analysis:** Thokozani Zungu, Shaffi Mdala, Halima Sumayya Twabi.

**Methodology:** Thokozani Zungu, Shaffi Mdala, Chatonda Manda, Petros Kayange.

**Supervision:** Shaffi Mdala, Chatonda Manda, Petros Kayange.

**Writing – original draft:** Thokozani Zungu, Shaffi Mdala, Chatonda Manda, Petros Kayange.

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
