## [Decision Letter · Decision Letter 0]

7 Sep 2020

PONE-D-20-22754

Characteristics and visual outcome of ocular trauma patients at Queen Elizabeth Central Hospital in Malawi

PLOS ONE

Dear Dr. Zungu,

Thank you for submitting your manuscript to PLOS ONE. After careful consideration, we feel that it has merit but does not fully meet PLOS ONE’s publication criteria as it currently stands. Therefore, we invite you to submit a revised version of the manuscript that addresses the points raised during the review process.

Three reviewers assessed your manuscript and their comments are attached.  Please submit your revision after addressing each of the comments.  All the comments are equally important. However, methodologically I would like to emphasize the comment made by one reviewer about the short followup.  This may lead to biased results.  The short followup should be explained.  The ethical comment about consent is also crucial.

We look forward to receiving your revised manuscript.

Kind regards,

Itamar Ashkenazi

Academic Editor

PLOS ONE

Journal Requirements:

Reviewers' comments:

Reviewer's Responses to Questions

**Comments to the Author**

1. Is the manuscript technically sound, and do the data support the conclusions?

Reviewer #1: Yes

Reviewer #2: Yes

Reviewer #3: Partly

2. Has the statistical analysis been performed appropriately and rigorously? 

Reviewer #1: Yes

Reviewer #2: Yes

Reviewer #3: Yes

3. Have the authors made all data underlying the findings in their manuscript fully available?

Reviewer #1: Yes

Reviewer #2: Yes

Reviewer #3: Yes

4. Is the manuscript presented in an intelligible fashion and written in standard English?

Reviewer #1: Yes

Reviewer #2: Yes

Reviewer #3: Yes

5. Review Comments to the Author

Reviewer #1: The authors present a study from Malawi reporting on incidence of serious eye injuries seen in a tertiary care center.

Overall, it is an interesting study and worthwhile examining and could help to educate current Government or other officials in this country to learn on the necessity of education and prevention to avoid loss of vision. However, the reviewer believes that the study could further improve and that there are some issues that should be addressed before publication.

The major criticisms are the short observation period and short follow-up.

Why is it not possible to continue data collection for nine more months to report at least one year?

Furthermore, the short follow-up of only 8 weeks leads to misleading conclusions re blind eyes after injury. The authors report that 25.3% of the affected eyes are blind. This statement needs to be toned down. E.g., remaining vitreous haemorrhage could lead to a „blind“ eye, but could eventually be treated by surgery.

what were the reasons for "blindness"? Unrepaired retinal detachments, haemorrhage, infections, other causes?

The study does not report any kind of intervention. How were eyes treated? How were open globe injuries treated? Did these eyes get one or more surgical interventions? Any surgical intervention at all?

Is blindness a result from non-treatment?

As a suggestion, another conclusion (next to better education and better prevention, as stated by the authors) could be that an improved medical system in the field of ophthalmology is needed. This urgent need could be highlighted in the discussion.

Minor comments:

Has there been any relation between time of presentation and the distance to the hospital?

Please also provide medain/range of time between injury and hospital presentation.

The level of education is not really relevant if large parts of their patients are children under 10 or 11 years!

How was urban and rural origin defined?

It is recommended to further distinguish eyes with no light perception from eyes with light perception. Any chance to help those eyes with light perception?

Reviewer #2: In this manuscript, Zungu et al present clinical information relating the frequency and consequences of ocular trauma in Malawi. While small and limited to events in one hospital over 2.5 months, it fills a gap in the understanding of ocular trauma in this country and provides data that can be used to design interventions aimed at reducing preventable blindness in Sub-Saharan Africa.

Two questions should be answered to increase the potential impact of this work:

How do the numbers compare with the 1982 work? Did the authors detect any changes over these four decades?

Lines 189-195: While the increased risk of blindness in the rural population is worth addressing, the main cause may not be a lack of access to fast medical care – the blindness OR for those that delayed the visit to the hospital was 0.97, suggesting that on those with severe lacerations leading to blindness, little could be done to save vision. Was there any insight into the types of injuries? Were those related to their labor, or due to increased exposure to sharp objects or animals (wild and/or domesticated)

Reviewer #3: Please find the below specific comments:

1. It has to be clearly mentioned in the Materials and Methods section as to how patient consent was obtained.

2. The incidence of ocular trauma in children was extrapolated to be 2.3 per 100,000 population. This Reviewer does not believe that the sample size of the study is sufficient to extrapolate the rates to a large population.

3. In Table 2, what constitutes “organic material”?

4. The demographic information about education and occupation does not add any context to the manuscript since the data analysis in Table 5 does not account for injuries based on occupation/education. Table 5 may be improved by including such analysis.

6. PLOS authors have the option to publish the peer review history of their article (what does this mean?). If published, this will include your full peer review and any attached files.

Reviewer #1: No

Reviewer #2: **Yes: **Alejandro P Adam

Reviewer #3: No

---

## [Author Response · Author response to Decision Letter 0]

18 Dec 2020

PLOS ONE

18 DECEMBER 2020

Dear Editor,

Thank you for the reviewers’ feedback on the manuscript titled “Characteristics and visual outcome of ocular trauma patients at Queen Elizabeth Central Hospital in Malawi” by Thokozani Zungu, Faik Gelisken, Shaffi Mdala, Halima Twabi, Chatonda Manda, Petros Kayange and Markus Schwering.

Below are the responses to each of the points that were raised.

Reviewer #1

The authors present a study from Malawi reporting on incidence of serious eye injuries seen in a tertiary care center. Overall, it is an interesting study and worthwhile examining and could help to educate current Government or other officials in this country to learn on the necessity of education and prevention to avoid loss of vision. However, the reviewer believes that the study could further improve and that there are some issues that should be addressed before publication. 

Question: The major criticisms are the short observation period and short follow-up.

Why is it not possible to continue data collection for nine more months to report at least one year? Furthermore, the short follow-up of only 8 weeks leads to misleading conclusions re blind eyes after injury. The authors report that 25.3% of the affected eyes are blind. This statement needs to be toned down. E.g. remaining vitreous haemorrhage could lead to a “blind” eye, but could eventually be treated by surgery.

Answer: We share the opinion of the reviewer regarding the short follow-up. There are two main reasons for the cut-off follow-up visit of 2 months. First, the patients’ transport costs had to be supported, and the limited budget of the College could cover only the visits within the two months. In view of this sociodemographic background, it was not possible to plan long-term follow-ups. Nevertheless, we acknowledge the shortcomings of our manuscript covering only the short-term follow-up.

We also realize that the meaning of monocular blindness was not well-portrayed in the first manuscript we submitted. To clarify on this query, we have highlighted on the definition of monocular blindness which we used and the characteristics of patients which were categorized as being blind in one eye at 8 weeks. As can be appreciated from the Materials and methods section, line number 105 to 110, Page 7. These are patients for whom visual prognosis is poor due to lack of treatment options in the local context.

Question: what were the reasons for "blindness"? Unrepaired retinal detachments, haemorrhage, infections, other causes?

Answer: The reasons for the blindness at two months have been added to the results section. (table 5) on page 15. We also added a comment regarding this point in the discussion section, page number 21, line number 287 to 289.

Question: The study does not report any kind of intervention. How were eyes treated? How were open globe injuries treated? Did these eyes get one or more surgical interventions? Any surgical intervention at all? Is blindness a result from non-treatment?

Answer: We added following sentences into the section of results: “Over a third of the population were admitted for treatment (n=42 41.2%), of which 95% (n=40 of 42) underwent surgery. Cornea repair was the most frequent surgery (n=18 of 40 45%) followed by cataract surgery (n=12 of 40)“. Table 3 on page 13, outlines the rest of the detail on type of surgical intervention.

Question: As a suggestion, another conclusion (next to better education and better prevention, as stated by the authors) could be that an improved medical system in the field of ophthalmology is needed. This urgent need could be highlighted in the discussion.

Answer: We added the following sentence in the conclusion paragraph in discussion.

Page 22, line number 310-314: “Additionally, apart from the irreversible complications outline in this study that resulted into blindness, corneal scaring was one common complication. Though a permanently blinding complication in our setting due to lack of corneal subspecialty services and huge cost on external referral, our study highlights the need for improved ophthalmological services in the country including subspecilization” 

Minor comments:

Question: Has there been any relation between time of presentation and the distance to the hospital?

Answer: We added the following sentence in the results.

Page 15, line number 200 to 201: “…There was no association between time of presentation and the distance to the hospital (OR 1.01, p 0.006 (CI1.00-1.02) on univariate analysis..”.

Question: Please also provide medain/range of time between injury and hospital presentation.

Answer: This has been included in table 1 page 9,

Question: The level of education is not really relevant if large parts of their patients are children under 10 or 11 years!

Answer: we share the opinion of the review and this has been removed from the manuscript. Table 1, page 9.

Question: How was urban and rural origin defined?

Answer: We have added the following information in the methods.

‘‘In line with the Malawi National Statistics Office’s Third Integrated Household Survey (IHS3) 2010-2011, patients were classified residing in urban areas if they came from any of Malawi’s major cities Lilongwe City, Blantyre City, Mzuzu City, and the Municipality of Zomba. Patients coming from all other areas were considered as residing in rural areas‘‘ page 6, line number 89 to 93.

Question: It is recommended to further distinguish eyes with no light perception from eyes with light perception. Any chance to help those eyes with light perception?

Answer: We share the view of the reviewer. However visual acuity data in the present study was collected as categorized by the WHO classification of vision. We note this as limitation in our study. However, the blind eyes had limited potential due to lack of subspecialty treatment i.e. corneal transplant surgery and vitreoretinal surgery to restore vision.

Reviewer #2:

In this manuscript, Zungu et al present clinical information relating the frequency and consequences of ocular trauma in Malawi. While small and limited to events in one hospital over 2.5 months, it fills a gap in the understanding of ocular trauma in this country and provides data that can be used to design interventions aimed at reducing preventable blindness in Sub-Saharan Africa.

Two questions should be answered to increase the potential impact of this work:

Question:

How do the numbers compare with the 1982 work? Did the authors detect any changes over these four decades?

Answer: We added the following sentence in the section of discussion. Page 21, Line number 290-298

“The previous ocular trauma study done in Malawi over four decades ago highlighted common use of traditional eye medication prior to hospital presentation(8). However, this finding is not reported in the current study. There has been an increase in the number of people affected with ocular injuries, with 205 people in the past study over 2 years’ period while in the current study 102 patients presented in about 3 months’ period. The duration before presenting to the hospital has also considerably been reduced from 10 days in the previous study to 2 days in the present. Hence the present study reports a change in ocular trauma trends in the region with increasing number of ocular injuries”.

Question: Lines 189-195: While the increased risk of blindness in the rural population is worth addressing, the main cause may not be a lack of access to fast medical care – the blindness OR for those that delayed the visit to the hospital was 0.97, suggesting that on those with severe lacerations leading to blindness, little could be done to save vision. Was there any insight into the types of injuries? Were those related to their labor, or due to increased exposure to sharp objects or animals (wild and/or domesticated)

Answer: The present study highlights that living in rural areas and sustaining open globe injuries were associated with monocular blindness. However, occupation was not significantly associated with monocular blindness as shown on Table 6 page 17.

Reviewer #3: Please find the below specific comments:

1. It has to be clearly mentioned in the Materials and Methods section as to how patient consent was obtained.

Answer; We added the following sentence on page 6, line number 82-84.

‘‘Following administration of initial treatment on the day of presentation, ocular trauma patients attending the hospital were enrolled into the study by obtaining their written informed consent with the aid of a patient information sheet‘‘

2. The incidence of ocular trauma in children was extrapolated to be 2.3 per 100,000 population. This Reviewer does not believe that the sample size of the study is sufficient to extrapolate the rates to a large population.

Answer: We share this opinion and have thus withdrawn the following sentences from the manuscript.

‘‘Adding to this is a finding from this present study that the annual incidence of paediatric ocular trauma in Malawi has been extrapolated to be 2.3 cases per 100,000 population, a proportion that is three times higher than that reported in the UK.‘‘

3. In Table 2, what constitutes “organic material”?

Answer: the word organic had been replaced with vegetative page 10, line 155. 

“The vegetative materials included sticks, wood (n=38 37.3%).

4. The demographic information about education and occupation does not add any context to the manuscript since the data analysis in Table 5 does not account for injuries based on occupation/education. Table 5 may be improved by including such analysis.

Answer. We have factored in occupation in the table 6 on page 17. However, level of education was removed from the manuscript following guidance from reviewer 1

---

## [Decision Letter · Decision Letter 1]

15 Jan 2021

Characteristics and visual outcome of ocular trauma patients at Queen Elizabeth Central Hospital in Malawi

PONE-D-20-22754R1

Dear Dr. Zungu,

We’re pleased to inform you that your manuscript has been judged scientifically suitable for publication and will be formally accepted for publication once it meets all outstanding technical requirements.

Kind regards,

Itamar Ashkenazi

Academic Editor

PLOS ONE

Additional Editor Comments (optional):

Reviewers' comments:

Reviewer's Responses to Questions

**Comments to the Author**

1. If the authors have adequately addressed your comments raised in a previous round of review and you feel that this manuscript is now acceptable for publication, you may indicate that here to bypass the “Comments to the Author” section, enter your conflict of interest statement in the “Confidential to Editor” section, and submit your "Accept" recommendation.

Reviewer #2: All comments have been addressed

2. Is the manuscript technically sound, and do the data support the conclusions?

Reviewer #2: Yes

3. Has the statistical analysis been performed appropriately and rigorously? 

Reviewer #2: Yes

4. Have the authors made all data underlying the findings in their manuscript fully available?

Reviewer #2: Yes

5. Is the manuscript presented in an intelligible fashion and written in standard English?

Reviewer #2: Yes

6. Review Comments to the Author

Reviewer #2: (No Response)

7. PLOS authors have the option to publish the peer review history of their article (what does this mean?). If published, this will include your full peer review and any attached files.

Reviewer #2: No

---

## [Editor Report · Acceptance letter]

22 Jan 2021

PONE-D-20-22754R1 

Characteristics and visual outcome of ocular trauma patients at Queen Elizabeth Central Hospital in Malawi 

Dear Dr. Zungu:

I'm pleased to inform you that your manuscript has been deemed suitable for publication in PLOS ONE. Congratulations! Your manuscript is now with our production department. 

Kind regards, 

on behalf of

Dr. Itamar Ashkenazi 

Academic Editor

PLOS ONE